# A Practical Guide to Computational Tools for Engineering Biocatalytic Properties

**DOI:** 10.3390/ijms26030980

**Published:** 2025-01-24

**Authors:** Aitor Vega, Antoni Planas, Xevi Biarnés

**Affiliations:** 1Laboratory of Biochemistry, Institut Químic de Sarrià, Universitat Ramon Llull, Via Augusta 390, 08017 Barcelona, Spain; aitorvegas@iqs.url.edu; 2Royal Academy of Sciences and Arts of Barcelona, 08002 Barcelona, Spain

**Keywords:** computational protein engineering, enzyme design, computational prediction, molecular recognition, binding affinity, catalytic efficiency, protein stability, protein solubility, molecular modeling

## Abstract

The growing demand for efficient, selective, and stable enzymes has fueled advancements in computational enzyme engineering, a field that complements experimental methods to accelerate enzyme discovery. With a plethora of software and tools available, researchers from different disciplines often face challenges in selecting the most suitable method that meets their requirements and available starting data. This review categorizes the computational tools available for enzyme engineering based on their capacity to enhance the following specific biocatalytic properties of biotechnological interest: (i) protein–ligand affinity/selectivity, (ii) catalytic efficiency, (iii) thermostability, and (iv) solubility for recombinant enzyme production. By aligning tools with their respective scoring functions, we aim to guide researchers, particularly those new to computational methods, in selecting the appropriate software for the design of protein engineering campaigns. De novo enzyme design, involving the creation of novel proteins, is beyond this review’s scope. Instead, we focus on practical strategies for fine-tuning enzymatic performance within an established reference framework of natural proteins.

## 1. Introduction

The global protein engineering market size was valued at USD 2.16 billion in 2022 and is expected to grow at a compound annual growth rate (CAGR) of 16.60% from 2023 to 2030 [1]. The increasing demand for efficient, selective and stable enzymes at industrial and research levels during the past decade has contributed to the blooming of the computational protein engineering (CPE) field. Rational design using computational methods assists experimental approaches to speed up protein engineering campaigns aimed at developing novel biocatalysts.

With the increasing availability of software, web servers and methods for CPE [2,3,4,5], the decision on which of these methods to use has become a challenge for scientists from many different disciplines. Moreover, the selection of suitable CPE tools strongly depends on the degree of knowledge of the protein structure and the enzymatic mechanism [4].

Overall, CPE strategies consist of the following three differentiated steps [6]: (i) mutation selection (library design), (ii) mutant model generation (modeling software), and (iii) target-property evaluation of mutant enzyme, typically changes in stability, substrate affinity or reactivity. There exists a great availability of software and methods accounting for the three steps mentioned above, including some capable of performing more than one. However, selecting the appropriate software from this extensive range can result in a difficult decision task for researchers outside the computational biology field, especially because new methods are often presented as advancements in the theoretical framework they rely on, rather than on the property they optimize. In this sense, we think that the key parameters (criteria) driving the software/method selection, from a practical perspective, should be the biochemical properties to be optimized. This involves the classification/ranking of mutant enzymes according to the desired property. The process requires the definition of a suitable scoring function that measures and ranks the target property with great reliability and confidence. Enzymatic properties of biotechnological interest evaluated by the most common scoring functions include affinity/selectivity for a given substrate, catalytic efficiency, stability at different operation conditions (mainly thermostability and performance at specific pH) and solubility for the optimization of recombinant biocatalyst production and its application in homogenous systems. Following this criterion, the motivation of the present review is to group the available software, tools and strategies according to the measured biochemical property and categorize these groups by the methodology used to evaluate the scoring function. It is not our intention to exhaustively review the exact methodology behind the different software and scoring functions but to give insights into the method/theory they are based on and more importantly, their possible usage. The main objective of this review is to assist both dry and wet lab researchers—with no or hardly any experience using CPE tools—in the use of these scoring functions for protein engineering research. To simplify the extensive methodology, we grouped the CPE tools and scoring functions according to the following properties: (i) protein–ligand affinity, (ii) enzyme reactivity, (iii) thermostability and (iv) solubility (see Figure 1). Notice that de novo design is excluded from the classical CPE strategies. This approach is based on creating a new enzyme from scratch and thus, the strategies used are very different from those used for tuning the enzyme’s catalytic performance. In this sense, de novo protein design is out of the scope of this review.

## 2. Engineering Enzyme–Substrate Recognition

In CPE campaigns aimed at evolving biocatalysts, substrate binding affinity is the property that scientists try to modulate the most. Binding affinity can be directly correlated to enzymatic efficiency because it accounts for the molecular recognition of the substrate and its availability for the enzymatic reaction to take place. General methods to evaluate molecular recognition include computational docking and molecular dynamics, as well as machine learning. The most relevant computational tools to engineer enzyme–substrate recognition, grouped by methodology, are listed in Table 1.

### 2.1. Virtual Docking Tools to Assess and Predict Enzyme–Substrate Complexes

Molecular docking has been the most used computational method for improving protein–ligand binding affinity during the past two decades [47,48]. Docking is based on the generation of large ensembles of ligand poses followed by their evaluation by a suitable scoring function capable of ranking these binding modes. Molecular docking has traditionally been used for the virtual screening of ligand libraries in drug design. Nevertheless, docking can also be used for CPE in the same way as virtual screening; the difference is that instead of docking thousands of possible ligands upon the same receptor, hundreds or thousands of mutant protein structures are docked with the same target ligand/substrate. Structure-based docking may be a useful tool to annotate enzyme function [49].

There are many different approaches to computational docking, the key ingredient being the theoretical framework on which the ligand binding affinity is evaluated. Liu and Wang [50] made a remarkable effort by classifying the vast amount of scoring functions to evaluate the binding affinity present in the literature (more than 100 have been published in the past two decades according to their research). These can be classified into five categories: physics-based, empirical, semi-empirical, knowledge-based and descriptor-based scoring functions. We provide a selection of the most used scoring functions (Table 2) in the field of CPE. These are described together with the computational tool they are implemented in.

#### 2.1.1. Physics-Based Scoring Functions

These scoring schemes rely on force field-based molecular mechanics (MM), solvation models and even quantum mechanics (QM). Docking software using force field-based scoring functions such as DOCK [72] and GOLD [11] calculate the protein–ligand binding affinity summing up the van der Waals and electrostatic direct atom pairwise interactions. DOCK uses the AMBER force field [51], which contains parameters for the nonbonded interactions but does not include a parameter for the hydrogen bonds. DOCK has been extensively used in enzyme design for different purposes, including the identification of candidate enzymes for the construction of a synthetic pathway for Acetyl-CoA production [8] and deciphering herbicide-resistance mutations on Acetohydroxyacid synthase (AHAS) [9]. In this last example, DOCK 6.7 was used to dock herbicides into AHAS structures and then several hybrid methods were tested (e.g., MM, MM-PBSA, QM/MM-GBSA) for predicting the resistance-leading mutations. The MM-PBSA [73] method resulted in notably high accuracy in the prediction of mutants, experimentally altering herbicide resistance.

GOLD Docking Software [11] is included within CSD-Discovery software as a third-party application. GOLD is based on the Tripos force field [52], which lacks hydrogen-bonding terms but it is possible to overcome this lack in those cases where it is necessary with a hydrogen-bond term extracted from SYBYL 8.1 software (see empirical scoring functions).

ICM [36,74] Docking (MolSoft) uses an ECEPP/3 force field [53] with the addition of solvation-free energy and entropic contribution terms as the scoring functions. The software uses Monte Carlo-derived movements and minimization of interaction potentials for ligand pose generation. ICM has been widely used for the virtual screening of protein structures including the MOR42-3 receptor [37], LSD1 inhibitors [75] and the world’s largest virtual screening assay, in which 10 million chemical compounds were screened in 11 h, which led to the identification of three lead compounds [76].

#### 2.1.2. Empirical Scoring Functions

These are usually developed by linear regression using a training dataset to reduce the function to a linear equation accounting for only a few physicochemical descriptors. Discovery Studio [17] is a comprehensive suite of validated science applications built on BIOVIA Pipeline Pilot. It presents two sibling scoring functions named LigScore1 and LigScore2, both are linear equations based on three descriptors: van der Waals interaction, a polar attraction term and a desolvation penalty.

Chemscore and ChemPLP are empirical scoring functions provided by GOLD that are derived empirically from regression to experimentally determine the binding affinities of protein–ligand complexes [54]. GOLD has been successfully used for inverse virtual screening [77], lead optimization [16] and identifying the correct binding mode of molecules [78].

FlexX [25] is one of the most used docking software historically. It presents an automatic method for docking organic ligands using an empirical scoring function derived from Böhm’s work [79]. The scoring function accounts for H-bonds, ionic interactions, aromatic groups interactions, lipophilic contact and the number of rotatable bonds, all of them being adjustable parameters except for the number of rotatable bonds. FlexX scoring function also presents a scaling function penalizing deviations from the ideal geometry. FlexX4 performs incredibly fast in virtual screening assays [80,81].

Other less extended virtual docking tools that implement empirical scoring functions are Surflex and MolDcok (Molegro Virtual Docker). Surflex [55] is a fully automatic flexible molecular docking algorithm that provides a scoring function able to effectively model protein–ligand noncovalent interactions. The scoring function is the sum of hydrophobic and polar complementarity (dominant terms), entropic and solvation terms. Molegro Virtual Docker [56] (MVD) scoring function is based on piecewise linear potential (PLP) extended with an H-bond directionality term. MVD was tested using the GOLD dataset [82] for docking accuracy. The results showed better performance for MVD (87% accuracy) [56] than GLIDE, GOLD, FlexX and SurfleX. MVD presents an easy-to-use interface.

#### 2.1.3. Semi-Empirical Scoring Functions

The most popular and versatile scoring functions are those that combine both physical terms (force fields) and experimentally derived parameters (empirical scores). These are known as semi-empirical scoring functions.

GLIDE [20] (Schrödinger) uses GlideScore 2.5, a semi-empirical scoring function that combines empirical and force field-based terms. It presents two distinct scores derived from the ChemScore function: Standard-Precision (SP) and Extra-Precision (XP) Glide. SP can be used in virtual screening assays to minimize false negatives, as it is more tolerant than the XP version. Oppositely, the XP score is a stricter function presenting severe penalties for poses violating physical chemistry principles, thus minimizing false positives.

GOLD also implements force field-based scoring functions complemented with empirical terms (termed semi-empirical scoring functions). The GoldScore scoring function presents an H-bond term, a pairwise dispersion potential describing contribution to the hydrophobic energy of binding and an MM term accounting for the ligand’s internal energy.

AutoDock4 [29] is the last version of AutoDock [83] original software, the crown jewel among docking software with more than 10 thousand citations to date. The high applicability of AutoDock relies on the continuous upgrade and addition of new functionalities in each released version. Autodock4 scoring function [58] uses a semi-empirical force field including an improved thermodynamic model of the binding process, an empirical method to estimate surrounding water contribution and a full desolvation model. The presence of the thermodynamic model allows the incorporation of moderate protein flexibility and the use of the scoring function for protein–protein docking. AutoDock4 provides high-quality predictions of ligand conformations and good correlations between predicted inhibition constants and experimental data [84]. In addition to Autodock4 and AutodockTools [29], Autodock Vina [30] is two orders of magnitude faster than Autodock4 because it automatically calculates grid maps, and it is better at finding accurate binding poses [84]. Vina scoring function is very similar to the X-Score [57] scoring function (Xtool v1.2 software) except for including intramolecular contributions apart from intermolecular. Both scoring functions are considered empirical but they were also calibrated using experimental affinity measurements from the PDBbind database. X-Score scoring is commonly a combination of three individual scoring functions named HPScore, HMScore and HSScore, and it is possible for the user to modify this combination. Nevertheless, X-score does not perform molecular docking by itself, so it is necessary to apply it combined with a molecular docking program as a generator of binding poses. X-score can be used to re-rank binding poses from other docking software.

Autodock and GOLD use the Lamarckian genetic algorithm during the pose prediction procedure, which highly increases the computational cost. On the other hand, GLIDE uses anchor and growth strategies to reduce time costs and make them affordable for virtual screening assays.

#### 2.1.4. Knowledge-Based Scoring Functions

These are derived from statistical information of frequently observed intermolecular close contacts by using the Potential of Mean Force (PMF [59] principle defined by the inverse Boltzmann relation [85].

The PMF [59] score is a measure of the binding free energy of a protein–ligand complex calculated as the sum of the atom pair interactions as a function of distance derived from the Brookhaven PDB database. A revisited version of PMF (PMF04 [60]) was derived from 7152 protein–ligand complexes, a 10-fold increase compared to the original PMF score (PMF99) derived from 697 complexes.

The Drugscore^PPI^ web server [62] performs alanine scanning for protein–protein interactions. The scoring function includes distance-dependent pair-potentials derived from 851 complex structures and 309 experimental results from alanine scanning.

Other less extended virtual docking tools that implement knowledge-based scoring functions are FRED, HYBRID and POSIT implemented in OEDocking (OpenEye [40]. FRED is a docking software that uses multiple knowledge-based scoring functions at different stages during exhaustive search including Shapegauss, PLP, Chemgauss, Chemscore, Screenscore, Chemical Gaussian Overlay (CGO), and Chemical Gaussian Tanimoto (CGT). However, the default scoring function used in FRED to rank poses is Chemgauss, which uses a Gaussian-smoothed potential [61] for measuring ligand pose complementarity to the active site. One successful application of FRED is the discovery of BChE inhibitors at the nano-molar range [41]. Similarly, HYBRID makes use of the same scoring functions as FRED except during the exhaustive search, where it uses the CGO ligand-based scoring function. Interestingly, HYBRID allows for the use of multiple conformations of the protein where the best structure is selected and used based on the docking database. Finally, POSIT [86] is a ligand-guided docking method that uses existing information about bound ligands for improving pose prediction. Interestingly, POSIT can determine the best-suited protein structure when provided with multiple structures, a potential application for CPE.

#### 2.1.5. Machine Learning-Based Scoring Functions

These scoring functions are based on a nonlinear fitting of the experimental measures of binding affinity and the features describing the protein–ligand complex. These scoring functions tend to perform better than previously described scoring systems given that linear fitting may not describe the real relationship of the score with the binding affinity. In addition, the computational cost of using ML algorithms for protein design is extremely reduced in comparison to using traditional rational design methods. However, the accuracy of an ML scoring function strongly relies on the training dataset used. In this sense, there exists a vast number of available datasets to benchmark binding affinity results derived from experimental studies of mutant proteins. The PDBbind Database [87] is the most widely used with binding affinity data for 19,588 complexes or compounds. Other examples of available databases are BindingDB [88], PubChem [89] and ChEMBL [90], which are commonly used for developing scoring functions based on specific protein–ligand complex targets, whereas the DUD [91], DUD-E [92] and MUV [93] databases were designed for virtual screening purposes. However, using ML-derived scoring functions for engineering catalytic properties is challenging given the extremely large diversity of reaction types, mechanisms, cofactors, experimental reaction conditions, substrate specificities and promiscuities [94].

ML scoring functions for docking are based on different methods such as Random Forest, Support Vector Machine, Artificial Neural Networks and Deep Learning Neural Networks [95]. Random forests, or random decision forests [96] consist of a learning method that constructs a multitude of decision trees giving as an output a categorical/classification response (i.e., active or inactive) or an average continuous prediction (numerical value for regression scoring) of the individual trees. Importantly, Random Forest-based scoring functions have been seen to perform poorly for binding pose prediction and virtual screening [97] but still outperform classical scoring functions [98] (e.g., RF-Score-v3 [63], RF-Score-VS [64] and Δ_vina_RF_20_ [65]). An accurate prediction of the binding mode is necessary to use these scoring functions. An example of application is the use of Δ_vina_RF_20_ for deciphering enzyme promiscuity of cytochrome c in the formation of cytochrome C-cyclo [6]aramide binding complex which exhibited higher activities than unmodified cytochrome c in the oxidation of benzhydrol to benzophenone [99].

Support Vector Machines were originally designed for classification and have been mainly employed for discriminating active and non-active ligand poses. A derived method named support vector regression can be used for regression analysis [100]. These ML methods rely on supervised learning by using nonlinear kernel functions, which can describe the covariance structure of the fitness landscape similarly to Gaussian processing [101]. Examples of this type of scoring functions are SVMGen [66] (classification of protein kinases), ID-Score [67] (used for re-scoring and predicting the sensitivity spectrum of various serine hydrolases to OP pesticides [102]) and PLEIC-SVM [68] (virtual screening of protein kinases, proteases and GPCR outperforming GLIDE predictions).

Artificial Neural Networks consists of a simulation of a brain functioning model with neurons organized as layers. DDFA [69], BgN-Score/BsN-Score [70] and NNScore 2.0 [71] are examples of these types of scoring functions for docking. DDFA (docking data feature analysis) was developed for virtual screening and re-ranking purposes with minimal extra computing time. It consists of five types of features derived from Autodock, Vina and Rosetta Ligand showing a similar performance to other docking software such as ICM, Vina and Glide using DUD dataset [71].

Overall, these ML approaches have been used in drug discovery, but we foresee the use of this type of scoring functions for the engineering of substrate specificity by coupling them with experimental library generation methods, binding affinity measurements and re-scoring the most promising binding poses. Moreover, these methods are recommended to be used for a more precise description of substrate true binding pose and the subsequent analysis of a more accurate pharmacophore for rational design of the binding pocket.

Su et al. performed a comparative assessment of scoring functions using a set of 285 protein–ligand complexes. This study revealed that VinaRF_20_ (a Random Forest-tuned version of Vina) [65] performs the best for computing binding score correlated with experimental binding constants (scoring power). The work of Su et al. revealed that scoring functions such as those implemented in X-Score, ChemPLP (GOLD), ChemScore (SYBYL) and Discovery Studio perform very well in terms of scoring power while London-dG (MOE), PMF (SYBYL) and PMF04 (Discovery Studio) decrease in performance. When assessing the ranking power (the ability to rank correctly known ligands), VinaRF_20_ performs the best again and ChemPLP (GOLD), DrugScore (CSD), LigScore (Discovery Studio) and X-score also correlate very well the ranking of known ligands.

### 2.2. Examples of Substrate Specificity Engineering Using Virtual Docking

One of the main issues when it comes to in silico enhancing substrate specificity is the generation of multiple mutant protein structures on which to measure binding affinity of the targeted ligand and compare it to WT. When the enzymatic reaction is well known and established, there are two main approaches for computationally measuring mutational effects on protein–ligand affinity/selectivity. The first approach is to rationally propose mutations by visually inspecting protein–ligand complex [23,38]. Usually this is performed using crystallographic structures of WT proteins or mutants (if available) or with homology models. By docking target substrate and refining binding poses with a suitable scoring function, a pharmacophore can be established, and then rational mutations are proposed (rational design). The second approach involves the generation of a smart or exhaustive mutant protein library to measure the affinity and/or selectivity for a given substrate with a precise orientation.

#### 2.2.1. Engineering of a Lipase for Omega-3 Fatty Acid Selectivity [32]

Two different approaches were tested for improving fatty acid selectivity (using two substrates, EPA and DHA) of a *Geobacillus thermovalorans* Lipase (GTL), rational and semi-rational design. Rational design involved the identification of binding-responsible residues present at one of the four pockets (acyl binding-site) based on the literature. V171 and L183 were replaced by bulkier amino acids resulting in a double mutant V171L/L183F, named DM-GTL. A two-step approximation was used consisting of (i) calculating the best binding mode presenting reactive orientation and (ii) calculating binding energy with no further flexibility by activating Autodock Vina “score-only” option. Semi-rational design consisted of identifying interacting residue for both substrates by virtual docking using Autodock Vina, with flexible side chain method. The 10 best binding modes for each substrate were analyzed. Most frequently observed (at least 15 out of 20 modes) to interact with the substrates were important for substrate selectivity. Amino acids playing an important role in the activity of the enzyme were excluded and the remaining positions were selected for site saturation mutagenesis (SSM). Six positions were screened: 170, 171, 244, 319, 358 and 359. From these six positions, 960 individual mutants were generated where 210 colonies showed lipolytic activity and, from these, 28 showed improved fatty acid hydrolysis. After sequencing, 13 different mutations were revealed at positions 170, 171 and 359.

#### 2.2.2. Rational Re-Design of Candida Antarctica Lipase B (CALB) Towards Diels–Alder Activity [13]

Lipases are versatile biocatalysits commonly used to catalyze a myriad of organic reactions. *Candida Antarctica* Lipase B (CALB) was engineered for a Diels–Alder reaction. Six enzyme variants of CALB were considered: WT, S105A, I189A, S105A/I189A, I285A and S105A/I285A. These variants were selected based on previous studies and insights after visual inspection. CALB PDB structure (1LBT) was used for building homology models with SwissPDB. GOLD suite (Genetic Algorithm) was used to generate different ligand poses on each variant and were evaluated with ChemScore scoring function (H-bond restrictions between dienophile carbonyl oxygen and T40-NH, T40-OỿH and Q106-NH). In total, 50 different poses were generated for each molecule and were scored using composite scoring function. After numerical and visual analysis, the best poses were saved for docking of the diene. Enzyme variants were MD-relaxed and docked in the same way as non-relaxed structures. Near attack conformation analysis (NAC) and DFT calculations were used to elucidate activation energy barriers. Results showed that S105A/I189A variant gives up to 5% ‘loose’ NAC geometries in acetonitrile for one given simulation, and even higher in water (8.5%). The redesigned enzyme showed effective Diels–Alder activity in accordance with these computational results.

#### 2.2.3. Identification and Engineering of the Key Residues at the Crevice-like Binding Site of Lipases Responsible for Activity and Substrate Specificity [34]

The process of molecular docking was performed by AutoDock 4.0 to explore the binding space of the enzyme–substrate complex. Flexible docking was carried out to evaluate ligand binding energies over the conformations search space using the Lamarckian genetic algorithm. The synergistic effects between Phe207 and Phe259 led to higher activity of the double mutant P207F/L259F than that of the single mutants. Amino acid residues located at the crevice-like binding sites of four representative lipases were rationally engineered and the obtained double mutants exhibited significantly improved activity towards p-nitrophenyl esters.

#### 2.2.4. Constructing a Synthetic Pathway for Acetyl-Coenzyme a from One-Carbon Through Enzyme Design [8] (Exhaustive Library Search)

A synthetic Acetyl-CoA (SACA) pathway was constructed by repurposing a glycolaldehyde synthase and an acetyl-phosphate synthase. First, a theozyme was constructed including thiamine diphosphate (ThDP), glycoladehyde and glutamic acid (acid/base). A PDB search using this theozyme resulted in the identification of 37 non-redundant protein structures including the ThDP ligand. Then, the distance between C2 and glycolaldehyde was computed, as this distance plays a critical role for the catalytic reaction using DOCK 6 software. The docking procedure revealed six enzyme candidates presenting short distances with clear function annotations. Experimentally, three of the six candidates exhibited desired reaction activity and were selected for a directed evolution study. The engineered glycolaldehyde synthase exhibited more than 70-fold increased catalytic activity.

#### 2.2.5. Creating Space for Large Acceptors: Rational Biocatalyst Design for Resveratrol Glycosylation in an Aqueous System [48]

Polyphenols display several interesting properties but their low solubility limits practical applications. Sucrose phosphorylase (SP) can produce α-glucosides through a transglycosylation reaction with sucrose as donor substrate. Glycosylation of resveratrol can dramatically improve its solubility and bioavailability. However, resveratrol binding to SP is hindered by an active-site loop, according to docking and modeling studies. Indeed, the unbolted loop variant R134A showed useful affinity for resveratrol (Km = 185 mM) and could be used for the quantitative production of resveratrol 3-α-glucoside in an aqueous system. In silico mutagenesis and docking studies indeed indicated that substitution of R134 with smaller residues (e.g., alanine) would leave an opening in the enzyme’s closed conformation, enabling the second ring of resveratrol to be accommodated.

### 2.3. Machine-Learning Tools for In Silico Enzyme Engineering

Shen et al. thoroughly reviewed the topic of Machine-Learning (ML) developments for protein–ligand docking [100]. Moreover, Mazurenko et al. also reviewed ML methods and databases for enzyme engineering [94]. In this section, we give a comprehensive overview of the available tools and databases as well as remarkable examples of ML use for biocatalyst design.

Similarly to QSAR models for lead optimization, ProSAR (Protein Sequence Activity Relationship) models can be used to infer the contributions of mutational effects on protein function coupled with efficient and minimal mutational experimental data [44]. Fox [42] developed a partial least-squares (PLS) regression ML methodology using a genetic algorithm (GA) for the directed evolution of proteins, which has been a major inspiration for other ProSAR models [44,45]. Interestingly, these models are sequence-based and do not need a three-dimensional structure assuming that phenotypical information is encoded at the protein sequence. This method was further developed with Halohydrin dehalogenase for improving the volumetric productivity of ethyl (R)-4-cyano-3-hydroxybutyrate(HN) [43] up to 4000-fold with respect to WT. In this case, after 18 ProSAR-driven iterative cycles of directed evolution and subsequent HTS activity assay, a 99.9% HN R-enantiomer was obtained with 99.5% purity. It was demonstrated that using ProSAR approximation can be useful for individual or multi-objective engineering of biocatalyst properties such as enantioselectivity, activity, thermostability and others. It is worth noting that only additive effects were considered when formulating the equation correlating mutations with enzyme function and so, only linear terms were computed. If necessary, other nonlinear terms can be added to account for synergic mutation interactions. Following ProSAR models aiding enzyme selectivity engineering, Berland et al. developed a web tool for the rational screening of mutant libraries using ProSAR [44] based on the previously established Fox strategy for ProSAR model building. This method was successfully tested for the engineering of (i) dextransucrase synthetic specificity towards α(1 → 3) or α(1 → 6) linkages in polysaccharide products and (ii) cytochrome P450 thermostability. In both cases, the model was demonstrated to be reliable enough to enable the prediction of new sequences: R^2^ = 0.60 for the dextransucrase and R^2^ = 0.94 for the cytochrome P450. Later, Berland and co-workers used this same strategy for the engineering of a transglucosylase for the production of kojibiose with controlled selectivity [45]. The semi-rational mutagenesis strategy resulted in a double mutant (L341I/Q345S) with 95% selectivity for kojibiose production and final purity of >99.5%.

Another example of ML methods successfully applied to specificity engineering is the GT-predict [46] tool for the identification of Glycosyl Transferase Superfamily 1 (GT1) potential novel substrates and functional annotation of uncharacterized GT1 members. The method is based on a decision tree approach trained on a varied combination of physicochemical properties and structural parameters. Its use in conjunction with structural approaches allowed for the identification of possibly important structural motifs and their roles within active sites. However, this method required a small but broad dataset of GT1’s activity performance on different substrates.

In order to reduce the necessity of a large dataset, Duan and Sun [103] developed an ML workflow to generate mutant libraries with a high enrichment ratio for the recognition of specific substrates using *M. jannaschii* tyrosyl-tRNA synthetase (TyrRS). In this case, the use of Rosetta modeling in combination with target-specific scoring function and ML (lightGBM) model calibration, the library enrichment ratio was increased by 11-fold compared with random mutation. By using the Rosetta EnzymeDesign method (de novo) to model the backbone changes and amino acid side chain packing upon reported mutations, they were able to predict the binding specificity of unnatural amino acids for every TyrRS mutant pair complex. The results showed that D158G/P mutants strongly influence backbone disruption of the α-helix at residues 158–163, opening the pocket to accommodate bulky unnatural amino acid.

## 3. Optimization of the Catalytic Efficiency of Enzymes

Simulating and numerically predicting enzymatic reactivity is a complex multi-objective challenge because it depends on different properties such as substrate binding selectivity [104], electrostatic environment (redox potential and electron transfer [104,105]), pocket hydrophobicity, and even substrate surface diffusion [106]. QSAR/QSPR (Quantitative structure−activity/property relationships) is one of the most common prediction models used for computational catalyst design. These models try to correlate hundreds of descriptors of the catalytic reaction with target properties to modulate, such as reactivity and selectivity [21,107]. Mainly, these models are built using regression analysis or ML methods for the description of a chemical space region accommodating the reaction. However, building QSAR/QSPR models for specific reactions requires high expertise and thus, its use is out of the scope of this review. Moreover, the use of QSAR models is not especially suitable for the automated design of catalysts. Several examples of QSAR have been described [108].

Computational reactivity modeling and prediction are extremely challenging given the high complexity of the electronic structure of the catalyst, as well as the conformational and configurational landscapes of the reaction’s transition state. Several protocols have been designed based on different simulation approaches, such as MD, QM and hybrid MM/QM [109]. In terms of scoring function for predicting enzymatic reactivity, there exist multiple approaches on how to measure it, ranging from protein–ligand geometric conformation favoring reaction, electron transfer probability and electron density prediction. Geometrical approaches rely on the prediction of the enzyme–substrate complex structure and measuring angles or distances between the designated reactant atoms fulfilling substrate catalytic requirements. The Empirical Valence Bond (EVB) theory calculates the reaction-free energies in the condensed phase. It uses potential surfaces for calculating the probability of electron transfer, using a calibrated Hamiltonian (operator corresponding to the total energy of the system in QM). The use of a Hamiltonian allows the approximation of the potential energy surface of a given reaction. To simplify, the catalytic reaction is modeled using two states corresponding to the reactants and products. Thus, EVB requires the reaction mechanism to be well characterized. Density-Functional Theory (DFT) is a QM modeling method capable of calculating electronic structure by means of functionals of the spatially dependent electron density. Although DFT calculations are sometimes unaffordable, given the necessary timescale of the simulations, their use in enzyme engineering has risen in recent years [110]. The combination of DFT + MD is a promising strategy to study structure and reactions [111]. The benchmark case of the example of citrate synthase [110] illustrates the applicability of DFT for engineering enzyme reactivity. However, the use of QM methods for engineering enzyme reactivity still requires great handling expertise and careful system model building [112]. In this section, we give an overview of protocols and methods successfully developed and applied to engineer enzyme catalytic efficiency in a comprehensive way. A list of the common computational tools and selected applications is collected in Table 3.

### 3.1. Computational Methods to Engineer the Catalytic Efficiency of Enzymes

The CASCO [113,115] protocol (CAtalytic Selectivity by COmputational design) uses high-throughput-multiple independent MD (HTMI-MD) simulations to engineer Limonene epoxide hydrolase enantioselective transformation of cyclopentene oxide [115], making it possible to replace experimental assays. This approach involves the design of a mutant enzyme with RosettaDesign [127] for the identification of low energetic structures. The scoring function approximating the enzyme’s reactivity consists of measuring the fraction of time of the MD simulation that the complex presents Transition State-like structures (pro-RR or pro-SS). Mutant structures are evaluated in terms of Near Attack Conformations (NAC), which satisfy geometrical-based restraints, such as the angle of nucleophilic attack and the distance between reactant atoms. This protocol also allows approximating protein–ligand binding affinity by measuring the ratio of NAC frequencies for each enantiomer. The use of HTMI-MD (ultra-short simulations) allows for increasing the protein conformational search space by screening thousands of Rosetta Design mutants while reducing computational cost, as demonstrated with epoxide hydrolase [115].

Houk et al. [119] used Density-Functional Theory (DFT) calculations and subsequent MD simulations to study the substrate binding mechanism of P450 monooxygenase. DFT calculations approximate electron density by means of QM (theozyme), which allows for the prediction of enzymatic site selectivity. The MD simulations (0.5 µs) were compared to the ideal geometry (H-O distance and O-H-C angles) of the stabilized TS via DFT calculations in order to propose rational mutations. Previously, Houk et al. had already used the DFT calculation for the design and optimization of a new dirhodium catalyst with high enantioselectivity [120] for the most accessible primary C-H bond by using ONIOM calculations. In this case, the “inside-out” protocol was used, which already had been applied to the so-called spiroligozymes. In this example, the protocol consisted of the de novo design of a transesterification catalyst and subsequently mutations improving its catalytic performance. Following QM calculations for computational reactivity, Cerqueira et al. reviewed these types of approaches and proposed a protocol based on catalytic geometry optimization [109]. The strategy includes locating the TS of the enzyme by generating intermediate structures of the catalytic pathway, which can be obtained by restraining one or more internal coordinates. Then, the potential energy surface is calculated for this ensemble of structures which allows for the determination of the TS.

Sherman’s group [21] combined MD simulations, docking and MM-GBSA scoring to approximate the catalytic reactivity of mutant enzymes. An MD simulation was used to generate an ensemble of bound configurations, which were scored by means of Induced Fit Docking (IFD) using GLIDE (GlideScore). IFD allows to account for protein flexibility while docking. In this case, the protocol was applied for the optimization of a ω-aminotransferase, identifying mutations increasing reactivity up to 20–60-fold for an imagabalin precursor with respect to WT. The protocol allows the binary predictive classification of mutant enzymes as active or inactive. Moreover, they developed a tuned IFD protocol including multiple iterations to be able to filter poses based on a distance cutoff between reactive PMP amine and substrate ketone group, accounting for reactive poses.

Maranas et al. developed IPRO and OptZyme [122] (derived from the IPRO suite of programs), a computational procedure for the redesign of *E. coli* β-glucuronidase (GUS) towards the use of novel substrate pNP-Gal. The protocol allows enzyme redesign in those cases where the TS structure of the reaction is unknown. In this case, it makes use of QM calculations to approximate a TS analogue for the identification of the rate-limiting step of the reaction. The idea behind this approach is to design mutations that lower the TS analogue energetic barrier. Results validated the correlation of the Interaction Energy upon a substrate (IE_s_) with Km (R^2^ = 0.960) and the IE_TSA_ with kcat/KM (R^2^ = 0.864). Moreover, this procedure is particularly useful for systems where solute entropy is negligible. IEs were calculated using IPRO.

The IPRO [124,125,128] suite of programs has been extensively used for different enzyme redesign purposes. It incorporates OptZyme (improvement of catalytic properties), OptGraft (design of the novel binding site) and OptCDR (antibody novel-complementarity design). The core functionality of IPRO is to randomly perturb the protein’s backbone around mutated residues for the identification of a new design with lower binding energy than the WT enzyme based on Interaction Energy calculations. IPRO allows for an iterative search of the mutations enhancing enzymatic activity/specificity. (Currently, IPRO only supports the use of the CHARMM force field). IPRO requires users to provide extensive information on how to run the experiment.

PELE [116] combines a Monte Carlo stochastic algorithm using a localized steered perturbation with side-chain prediction and energy minimization based on Metropolis acceptance/rejection criteria. The acceptance criteria ensure that perturbation does not lead further along the coordinates of a given reaction and/or large interaction potential energy increase, resulting in a series of local minima with a high structural correlation. This approach enables a large sampling of configurational space and thus permits efficient CPE towards target-property. The scoring function used for ranking is an OPLS-AA force field in which only the ligand and the backbone of the protein are considered. Desolvation effects are not considered in this case, which may be necessary for some CPE campaigns. PELE structure prediction capability reproduces long time scale processes efficiently reducing computational time–cost. In this way, PELE enables obtaining an atomic detailed mechanism of the protein–ligand-induced fit of its recognition process and of the ligand migration. PELE could also have been introduced in the previous section as a computational tool to optimize enzyme–substrate interactions. We present this method here given the possibility to perform single-point mutations mixed QM/MM calculations to update the charges of complex ligands or to obtain quick estimates of a biochemical reaction [111], which makes it useful for CPE. PELE was benchmarked by studying (i) aspirin binding to phospholipase A2 and Nuclear hormone receptors as a ligand refinement [117]. Example cases of CPE for reactivity enhancement are presented in the following section.

FuncLib [118] extends the PROSS protocol (see later) by designing stable networks of interacting residues within the active-site pocket of an enzyme aimed at increasing both protein stability and catalytic efficiency. Unlike other methods, FuncLib does not target specific substrates nor relies on models of enzymatic transition states. Rather, it exhaustively enumerates combinations of three to six mutations, and models each mutant using Rosetta. Designs are ranked by all-atom energy, prioritizing those that encode diverse stereochemical complementarities for alternative substrates, which do not need to be predefined. The method’s output is a repertoire of stable, highly efficient enzymes amenable to low-throughput experimental screening for desired activities, offering a practical solution for enzyme engineering and functional diversification. The generality of the methods was demonstrated by broadening the substrate selectivity of Acyl-CoA synthetases [118] towards larger aliphatic acids.

CADEE [123] (Computer-Aided Directed Evolution of Enzymes) is a computational framework used for the screening of thousands of enzyme variants based on the EVB approach. EVB can be used for large screening assays as it is fast and efficient, allowing us to obtain free energy calculations describing chemical reactivity in a physically meaningful way. CADEE requires a well-characterized system to obtain reliable results thus, a good quality EVB force field. CADEE can introduce mutations via alanine scanning. The CADEE framework was validated by comparing experimental results of Triosephosphate isomerase (*S. cerevisiae*) Kcat to calculated values of free energy, showing a correlation with activation free energies [123].

### 3.2. Examples of Enzymatic Reactivity Engineering Using Computational Methods

#### 3.2.1. Computational Design of Enantio-Complementary Epoxide Hydrolases for Asymmetric Synthesis of Aliphatic and Aromatic Diols (CASCO) [115]

Limonene epoxide–hydrolase substrate was docked in the active site and placed in a reactive configuration (NAC) using orientation/distance restraints. Next, the Rosetta Monte Carlo search algorithm was used to optimize side chain geometries of amino acids surrounding the active site for either pro-RR or pro-SS attack of the nucleophilic water on the epoxide carbon. A large number of parallel MD simulations with independently assigned initial atom velocities (HTMI-MD) were performed. The reactivity and selectivity of each mutant were predicted by scoring the fraction of snapshots in which the enzyme–substrate complex is in a pro-RR or pro-SS near-attack conformation (NAC).

#### 3.2.2. Insights into Laccase Engineering from Molecular Simulations: Toward a Binding-Focused Strategy (PELE) [104]

The objective of this study was to computationally design an evolved laccase with increased reactivity. *Pycnoporus cinnabarinus* laccase (PcL) and the substrates employed to screen activity were 2,2′-azino-bis(3-ethylbenzo- thiazoline-6-sulfonic acid) (ABTS) and 2,6-dimethoxyphenol (DMP). The CPE strategy is based on a combination of conformational sampling and quantum-chemical reactivity scoring based on changes in substrate’s spin density (electron transfer). The conformational space of the binding pocket is sampled using PELE. Subsequently, 20 mutant structures showing low binding energy poses were selected and their reactivity was scored by evaluating the amount of spin density localized on the substrate (evaluated using Mulliken partitioning method) with hybrid quantum mechanics−molecular mechanics (QM−MM) calculations. The QM region consisted of the substrate and residue’s first shell while the rest of the protein structure was treated with an OPLS-AA51 force field (classical MM). Desolvation effects were neglected to speed up calculations as the main objective was to screen large amounts of protein mutants with feasible computation time–cost. Mutant structure “hits” can be visualized as a bi-dimensional plot showing binding energy versus copper-substrate distance (substrate’s center of mass). Two different substrate-binding modes were used for the DMP substrate (resulting from docking studies using GLIDE). The evolved laccase carries five mutations: P394H and N208S, located in the T1 pocket, N331D and D341N, relatively close to the substrate entrance, and R280H, located far away on the protein surface for both substrates (kcat 13-fold improvement for ABTS and ~19-fold for DMP substrate). Correlation studies between the rate constant (kcat) with the redox potential difference (ΔE°) suggested the reduction is the rate-limiting step of the catalytic process determined by the free energy difference between products and reactants.

#### 3.2.3. Computational Redesign of Acyl-ACP Thioesterase with Improved Selectivity Towards Medium-Chain-Length Fatty Acids (IPRO) [126]

The IPRO algorithm was used to design thioesterase (TesA) variants with enhanced C12 or C8 specificity while maintaining high activity. After four rounds of structure-guided mutagenesis, we identified three variants with enhanced production (reactivity) of dodecanoic acid (C12) and 27 variants with enhanced production of octanoic acid (C8). The top variants reached up to 49% C12 and 50% C8 while exceeding native levels of total free fatty acids. The potential of the IPRO algorithm to aid in protein engineering efforts was demonstrated using a Design–Build–Test–Learn approach to alter the substrate preference of TesA.

## 4. Engineering Protein Stability

Protein folding is mainly driven by intramolecular interactions between residues and hydrophobic effects leading to a well-defined native protein structure [129]. However, the native conformation co-exists with misfolded and unfolded states. Free energy differences between the different conformational states of a protein determine which of the states is most populated. Protein conformational stability is thus defined as the free energy equilibrium between folded and misfolded states. Protein structures presenting lower energies in misfolded states can lead to aggregation.

Protein stability also refers to the resistance capacity of the protein’s native structure to high temperatures, denaturant agents, proteases, and non-physiological pH. The overall stability of a protein is determined by non-covalent interactions (e.g., hydrophobicity, van der Waals interactions, hydrogen bonding, and electrostatics) forming interaction networks that stabilize the native structure.

The topic of engineering biocatalysts for improved stability was profoundly reviewed by Bommarius and Paye [130] and more recently by Musil et al. [5] from a computational perspective. In the following sections, we present a selection of computational methods and scoring functions for the rational and automated computational design of biocatalysts with enhanced stability as well as presenting successful examples of use (see the list in Table 4).

### 4.1. Computational Methods to Engineer the Protein Stability

#### 4.1.1. Phylogenetic Analysis-Based Methods

Ancestral sequence reconstruction (ASR) is based on the assumption that ancestral enzymes existed in a much hotter environment billions of years ago with thermophilic organisms present on the earliest branches of the tree of life. In this way, searching for ancestral sequences using phylogenetic analysis must reveal thermostable enzymes. BAli-Phy [131] implements ASR for enzyme optimization. The application of this method to adenylate kinase (Adk) resulted in the improvement of thermostability at 35 °C and near 2-fold catalytic activity enhancement [132]. *E. coli* expression of ancestral and modern Adk sequences revealed salt bridges as the primary source for differential stability. In a similar manner, Damborsky et al. used ASR theory for the improvement of Haloalkane dehalogenase thermostability (∆Tm up to 24 °C) [134]. On the other hand, consensus design (CD) relies on the assumption that the consensus residue at a given position in a multiple sequence analysis must be contributing the most to protein stabilization (not considering catalytic residues) compared to non-conserved residues [156]. CD differs from ASR in the way that it does not try to reconstruct ancestral sequences but performs Multiple Sequence Alignment (MSA) to extract the most conserved residues rather than reconstructing the entire phylogeny. 

#### 4.1.2. Rational Design by Molecular Modeling

Rational design involves the study and characterization of the contribution of each residue to protein stability. One approximation is the modification [157] or even deletion [158] of flexible loops or residues, which can lead to the improvement of enzyme thermostability. A remarkable strategy is ‘loop grafting’, which stands for the accommodation/transfer of validated thermostable loops from other proteins to the target [159]. This strategy was successfully applied to enhance the thermostability of subtilisin E-S7 (SES7) peptidase [160] and proline 4-hydroxylase [161]. Other rational approaches for the enhancement of thermostability involve protein surface-charge optimization [162,163], mutation of surface residues following the proline rule [164,165] and the introduction of disulfide bonds [166,167,168]. Rational design is also applied for the enhancement of enzyme stability for detergent formulation, which is a major challenge in laundry industries. For example, *Bacillus stearothermophilus* neopullulanase [169] (bsNpl) was rationally engineered for improved activity at elevated temperatures and high surfactant concentrations. Protein structure was visually inspected for determination of internal cavities and residue positions for which an amino acid exchange could be beneficial. This rational design resulted in a drastic stabilization of bsNpl against inactivation by heat and detergents derived from five mutations. Importantly, the catalytic activity of the enzyme remained identical to the WT enzyme.

Available software for protein stability prediction based on Gibbs free energy calculation include FoldX [140], ERIS [138], PoPMuSiC [135]. FoldX is an empirical force field developed for the prediction of mutational effects on the stability, folding and dynamics of proteins. The force field consists of a linear combination of empirical terms, including non-bonded terms (H-bonds, VdW and electrostatics), solvent interactions accounting for (de)solvation effects and explicit treatment of water molecules with persistent interactions (more than two hydrogen bonds). A unique feature of FoldX among other force fields is the estimation of the entropy derived from statistical analysis of the phi–psi distribution of a given amino acid throughout a set of high-resolution crystal structures.

Another powerful tool for stability prediction is ERIS [138], a web server using a physical force field with atomic modeling and implemented backbone flexibility capabilities, allowing for higher predictive power on “small-to-large” mutations. The scoring function is expressed as a weighted sum of van der Waals forces, solvation, hydrogen bonding and backbone-dependent statistical energies. ERIS showed a correlation of 0.75 with experimental ∆∆G for 595 mutants of five proteins [138]. It also presents a pre-relaxation option for low-resolution structures; therefore, its use is recommended for homology modeling-derived protein structures.

PoPMuSiC [135] is a web server presented as a Protherm [170] subset, for the prediction of mutational effects on protein stability based on the use of statistical potentials (knowledge-based). It uses a force field equation based on 13 physical and biochemical terms, including amino acid type, solvent accessibility, torsion angles, backbone conformation and distance between geometric centers of the side chains for every pair of atoms. PopMuSic only requires the WT protein or peptide structure in a PDB format as an input.

#### 4.1.3. Knowledge-Based Scoring Functions

DFIRE [143] (Distance-scaled, Finite-Ideal gas REference state) is a knowledge-based potential for the prediction of folding stability. It is an all-atom, distance-dependent, pairwise statistical energy function used to calculate the Potential of Mean Force (PMF) for mutations with a decreased number of atoms (avoiding small-to-large mutation predictions). The predicted free energy change due to mutation is calculated by assuming no structural relaxation after mutations. An extension of DFIRE called dDFIRE (dipolar DFIRE) was developed by Yang and Zhou [129] based on the orientation angles involved in dipole–dipole interactions which significantly improved DFIRE performance in segment refolding. DFIRE has been successfully implemented into DMUTANT [144].

#### 4.1.4. Machine-Learning Methods and Scoring Functions

I-Mutant 3.0 [145] (an extension of I-Mutant 2.0 [171]) is a support vector machine (SVM)-based tool. It presents two different capabilities: (i) discrimination between stabilizing, destabilizing and neutral effects upon single point mutations and (ii) a regression estimator for predicting ∆∆G. I-Mutant 3.0 can use both protein sequence and structure with a prediction power of 56% and 61%, respectively, using data extracted from the Protherm database. The 3.0 version uses an input vector consisting of 42 values, including temperature, pH, residue type and residue environment. The last value accounts for the spatial environment when structure is available and for the nearest sequence neighbors when only using sequence data. On the other hand, MAESTRO [146] (multi-agent stability prediction upon point mutations) is a more complex ML-based software for protein stability prediction. MAESTRO is structure-based and was also trained using data from the Protherm [170] database. It combines neural networks with SVM, regression analysis and statistical potentials providing additional sequence and structural information (such as protein size or solvent accessibility) which can be used to select specific mutation sites. Individual results from the different agent predictors are combined in order to provide a consensus prediction for point mutations resulting in the multi-agent method. Moreover, MAESTRO software also presents running modes for disulfide-bond introduction-site prediction and multiple point mutation greedy scan. The results are presented as ∆∆G prediction with associated confidence estimation.

Deep learning methods have become increasingly prominent in enzyme engineering due to their ability to learn complex patterns from data to address not only protein stability but catalytic efficiency and substrate specificity as well. These are out of the scope of this review but have excellently been reviewed by many authors [94,172].

#### 4.1.5. Hybrid Approaches

Computational methods combining different theoretical frameworks also exist aimed at enhancing protein stability. PROSS [154,173] is a web server combining multiple sequence alignment analysis and Rossetta modeling for the calculation of energy differences upon single-point mutation to define a space of potentially stabilizing protein mutations. From these, the optimal combination of mutations is identified by combinatorial sequence design with Rosetta.

The FRESCO [152] (Framework for Rapid Enzyme Stabilization by Computational Libraries) strategy uses FoldX and Rosettaddg for the prediction of free energy ∆∆G derived from point mutations. It then uses the Dynamic Disulfide Discovery (DDD) algorithm (based on an ensemble of structures from an MD simulation) to search for the introduction of stabilizing disulfide bonds on limonene epoxide hydrolase. On the other hand, the use of the FRESCO strategy on glucose oxidase [142] was reported to enhance its thermostability by 8.5 °C with increased pH tolerance (up to pH 8) where the WT becomes inactive. Moreover, the combination of these stabilizing mutations resulted in a 2-fold activity increase for gluconic acid production at industrial viable conditions.

FireProt [153] is a web server combining energy- and evolution-based approaches for predicting highly stable multiple-point mutants. For the energy-based approach, FireProt performs a conservation and correlation analysis with subsequent filtering using Rosetta and FoldX predictions. On the other hand, the evolution-based approach performs back-to-consensus analysis and then uses FoldX for filtering.

### 4.2. Examples of Protein Stability Engineering Using Computational Methods

#### 4.2.1. Computation-Aided Engineering of Starch-Debranching Pullulanase from Bacillus Thermoleovorans for Enhanced Thermostability [141]

In this work, authors combined FoldX, DFIRE and I-Mutant 3.0 resulting in a 3.8 °C increased Tm and a 2.1-fold longer half-life than the wild type at 70 °C. First, FoldX was used to perform Site Saturation Mutagenesis on a list of MD-predicted flexible residues. Subsequently, DFIRE and I-Mutant 3.0 were used to verify predicted stable mutants. The procedure resulted in six experimentally confirmed mutants enhancing thermostability from 17 computational designs.

#### 4.2.2. Engineering a Thermostable Fungal GH10 Xylanase [137]

PoPMusic was used to predict potential key regions that might be crucial for enhancing Xyn10A_ASPNG thermostability. The feature of flexibility for each residue of the modeled Xyn10A was evaluated from the computation of protein folding free energy changes (−∆∆G) resulting from all possible amino acid substitutions. Four rounds of iterative saturation mutagenesis generated a quintuple mutant 4S1 (R25W/V29A/I31L/L43F/T58I) which exhibited thermal inactivation half-life (t_1/2_) at 60 °C that was prolonged by 30 folds in comparison with the wild-type enzyme. Furthermore, the mutant melting temperature (Tm) increased by 17.4 °C compared to the wild type. The notorious improvement of enzyme thermostability of 4S1 was attributed to the synergistic effects of the five mutations.

#### 4.2.3. Thermostability Improvement of the Glucose Oxidase from Aspergillus Niger for Efficient Gluconic Acid Production [142]

FRESCO workflow was used to design variants of a glucose oxidase from *Aspergillus niger* for industrial applications with minimal experimental screening. Energy calculations with FoldX, Rosetta_ddg and ABACUS were performed to identify the potentially stabilizing mutations for further evaluation. The relative folding free energy changes (ΔΔGFold) were predicted by the FoldX and Rosetta_ddg algorithms. To enrich the beneficial mutations in the in silico library, the mutations were subsequently screened by visual inspection and molecular dynamics (MD) simulation. For each mutant, five independent 100-ps MD simulations with different random set initial atom velocities were performed using the Yamber3 force field. The combined mutant AnGOD-m containing five stabilizing mutations (T10K, A36M, R145N, G274S and E374Q) showed a +8.5 °C higher Tm value compared to the wild-type enzyme. When the temperature was 40 °C, the variant maintained 85% residual activities at pH 5.5 and 6.0 and 75% residual activities at pH 7.0, while the wild type maintained approximately 75% residual activities at pH 5.5, and 60% residual activities at pH 6.0 and 7.0.

#### 4.2.4. Disulfide Bond Engineering of an Endoglucanase from Penicillium Verruculosum to Improve Its Thermostability [168]

A structure-based design of disulfide bonds was performed through Cys scanning to identify potential mutations that can result in disulfide bonds using Schrödinger’s BioLuminate software. Two improved enzyme variants, S127C-A165C (DSB2) and Y171C-L201C (DSB3), were obtained. Both engineered enzymes displayed a 15–21% increase in specific activity against carboxymethylcellulose and β-glucan compared to the wild-type. After incubation at 70 °C for 2 h, they retained 52–58% of their activity, while EGLII-wt retained only 38% of its activity.

## 5. Improving Protein Solubility

Protein solubility is a complex feature involving different physical and biological properties. Solubility is mainly related to the aggregation or self-association propensity of proteins which can be explained as an alternative and thermodynamically stable protein folding [174]. Solubility property can be described quantitatively by measuring protein expression (expression yield) or qualitatively (soluble/insoluble). There exists little knowledge about which descriptors can be used to predict protein solubility. It is known that negative surface charge correlates with increased solubility [175] and that protein aggregation is directly correlated to the number of aggregation-prone regions (APRs) present in protein sequences [174,176,177,178]. APRs are short 10–15 residue-long stretches in proteins with self-associate aggregation tendency into ordered intermolecular beta-sheet or “cross-beta” spines [179]. In fact, proteins have evolved to be soluble in native physiological culture conditions and as a result, recombinant protein over-expressed for industrial or therapeutic uses present high aggregation rates. Globular proteins present higher aggregation rates with approximately 2–4 APRs per domain [180] given their need for a hydrophobic core for secondary structure organization, thus generating aggregation-prone amino acid sequences. On the other hand, monoclonal antibodies (mAb) also present high aggregation rates promoted by APRs mainly located at complementary determining regions [181,182,183]. However, only solvent-accessible APRs can form stable interactions leading to protein aggregation. In contrast, buried APRs often contribute to protein structure and function. The disruption of these buried APRs without knowledge of their contributions can lead to protein destabilization and/or loss of function. Regarding negative surface charge driving aggregation, a key strategy for reducing the aggregation propensity is to modulate the isoelectric point [184], reducing the protein’s total net charge, which decreases the protein–protein repulsion and thus, increases integration chance probability.

### 5.1. Computational Methods to Improve Protein Solubility

Given the difficulty of obtaining other quantitative solubility measurements, no other measurable properties apart from APRs influencing solubility have been characterized to date [185]. In this sense, solubility prediction tools are mostly based on machine learning approaches, ranging from simple statistical approaches to modern nonlinear methods such as support vector machines, random forests, or deep neural networks for APR detection developed using available data [5]. For a more detailed review of the re-design of proteins for increased solubility, we recommend reviews by Navarro and Ventura [177,186].

The most common software tools for protein solubility design are listed in Table 5 grouped by the theoretical framework they rely on. Independent of the methodology, these tools can be divided into three categories: (i) tools based on the analysis of protein primary sequences, (ii) methods based on the evaluation of sequence solubility profiles and (iii) based on measuring the effect of point mutations into protein solubility. The first group comprises tools scoring protein sequences with a single value. Examples of these are SolubiS (APRs identification and stability prediction), ESPRESSO (expression and solubility estimation), Periscope (*E. coli* soluble expression in the periplasm), SoluProt (training dataset restricted to *E.coli* expression) and other tools trained using the TargetTrack [187] database such as SOLpro, PROSO II, ccSOL omics and DeepSol, which are very similar. The second group consists of tools scoring each protein residue with a single score indicating its contribution to the whole solubility of the protein. These tools can present aggregation-prone predictions or non-dimensional scores. Zyggregator and AGGRESCAN3D 2.0 are aggregation-prone predictive software tools while TANGO, WALTZ and PASTA 2.0 provide probability scores based on the training with amyloid-aggregate formation proteins. Finally, in the third group are those software tools specifically designed for measuring the effect of mutations on protein solubility. However, independently of the group, the outputs of these software tools are typically expressed as non-dimensional arbitrary scores with no correlation with measurable physical properties. Even so, generating quantitative scores for single-residue or fixed-size fragments is very useful for the rational design of soluble proteins, whereas whole-protein single solubility scores are useful for genomic projects [5]. Following this, the last group of software tools is the main object of discussion in this section as we intend to present a comprehensive review of software tools for CPE. The different software and methods presented in this section are divided according to the method they use for generating solubility prediction (Table 5): sequence analysis, structure analysis, machine learning or hybrid approaches.

#### 5.1.1. Sequence-Based Analysis

The GAP [188] (Generalized Aggregation Proneness) method is one of the first methods developed for the identification of APRs. It evaluates and classifies peptide sequences into amyloid fibril or amorphous β-aggregate-forming hexapeptides. The method relies on the observation that hexapeptides present position-specific amino acid propensities distinct from those of the amorphous β-aggregating. Although the GAP method takes protein sequence as input, it was developed using statistical analysis of computed frequencies of residue pair types from a dataset consisting of 139 amyloids and 168 amorphous peptides. The scoring function is a thermodynamic energy difference potential for each residue pair type (i,j) occurring at alternate and adjacent positions. First, residue pairs at nine different positions of hexapeptides were converted into propensity equations by normalization with overall residue pair composition in globular proteins. Then, these propensities were treated as partition functions and converted into thermodynamic energy potentials. The scoring function for predicting hexapeptides as fibril forming or amorphous β-aggregating is highly accurate for most of the peptides. Peptides showing high aggregation propensities are identified as APRs and thus, it is suitable for the rational design of mutant proteins with enhanced solubility.

TANGO [179] calculates the partition function of the conformational phase-space following the principle that any peptide segment can populate any of the structural states (β-turn, α-helix, β-sheet aggregation and α-helical aggregation) according to a Boltzmann distribution. This partition function is relative to the energy of each state for a given peptide segment and permits the identification of APRs from the protein primary sequence. TANGO results are normalized between 0 and 1 and presented as a non-dimensional score over protein segments, which was validated to have an aggregation prediction accuracy of 0.7 with experimental data [179]. TANGO incorporates four conformational states and different energy terms, considering hydrophobicity and solvation energetics, electrostatic interactions and hydrogen bonding. The model used by the TANGO algorithm is designed to predict β-aggregation in peptides and proteins and consists of a phase-space encompassing the random coil and the native conformations as well as other major conformational states, namely β-turn, α-helix and β-aggregate. Every segment of a peptide can populate each of these states according to a Boltzmann distribution. Therefore, to predict β-aggregating segments of a peptide, TANGO simply calculates the partition function of the phase space. TANGO is one of the first methods developed for computational prediction of protein aggregation and it has been widely applied stand-alone. For example, TANGO was used as the core algorithm for the design of anti-amyloid cyclic peptides with increased solubility for therapeutic use as an anti-Alzheimer treatment [190]. It was also applied for the identification of Src homology 2 (SH2) domain mutations increasing the yield of soluble TSAd-SH2 domains [191]. A 9-residue-long sequence (SAVTFVLTY) was identified as the key factor leading to beta-sheet aggregation. The TFV to GYT sequence mutation doubled the yield of soluble protein expression.

SODA [193] is a sequence-based method combining PASTA [195] aggregation energy score, ESpritz [211] intrinsic disorder propensity score, the negative Kyte–Doolittle hydrophobicity profile [212] and FESS [213] estimated secondary structure propensities for α-helix and β-strand. Each score difference ∆S is summed and normalized for the full sequence. SODA is very efficient at predicting solubility decreases with a prediction accuracy of 72%. SODA provides two types of analysis, namely ‘mutation mode’ and ‘full-protein mode’. The first provides the solubility change on sequence mutation. The second generates a profile describing the contribution to the solubility of each sequence position deduced from the effect of all possible mutations. SODA was used in the structural analysis of the STN1 gene involved in Coat Plus Syndrome for the solubility prediction of pathogenic mutations [194]. The results showed that 10 out of 30 pathogenic mutations decrease protein solubility. These findings are expected to be useful for the development of novel strategies for the therapeutic treatment of Coat Plus Syndrome.

PASTA 2.0 [195] is a web server that evaluates the stability of putative β-strand inter-molecular pairings between different sequence stretches and predicts amyloid formation regions from protein sequences. PASTA discriminates the orientation of β-strands between parallel and anti-parallel. The PASTA 2.0 scoring function was calibrated using the TESE dataset extending the energy parameters of the previous PASTA version (hydrogen bonding statistics on β-strands). In this version, an ML method was implemented for the detection of secondary structure while maintaining a disorder predictor. These two outputs provide structural information that is easy to interpret which can be related to aggregation prediction. PASTA 2.0 performance upon aggregation assignment (to a sequential stretch) was tested using an AMYLPRED2 test set consisting of 33 proteins with 1260 annotated aggregating residues. Results were compared to other aggregation predictors such as Aggrescan and TANGO. PASTA 2.0 showed 41% sensitivity with 85% in contrast with 35% and 14% of sensitivity for Aggrescan and TANGO. Moreover, PASTA 2.0 reached the highest MCC (Matthews correlation coefficient measuring the quality of binary classifications) with 0.24 in front of Aggrescan (0.13) and TANGO (0.14). These results allowed us to establish PASTA2.0 as a ‘hot spot’ predictor with high confidence, Triosephosphate isomerase [196] and Human helicase [197] being examples of use. In the first case, eight regions from human triosephosphate isomerase (HsTPI) were predicted as fibrillogenic-forming regions—with five of them located at β-strands regions—and were selected for experimental aggregation studies. Among these regions, four were experimentally validated as fibrillogenic, corresponding to the β3, β6, β7 and α8 of the TIM barrel. Likewise, PASTA 2.0 was used along with Aggrescan and FoldAmyloid for the prediction of aggregation regions in RNA Polymerase Interacting Helicase (HelD) from *B. subtilis* where 20 regions spread across sequence were suggested to aid the formation of amyloid-like fibrils.

ESPRESSO [198] (EStimation of PRotein ExpreSsion and Solubility) is a web server developed for the estimation of protein expression and solubility in *E. coli* and wheat germ. The training datasets consisted of 5100 proteins (1774 soluble and 3326 insoluble) and 2939 (1941 soluble and 998 insoluble) for *E. coli* and wheat germ, respectively. For the *E. coli* dataset, SVM was used for training statistical models for expression and solubility prediction, while a sequence pattern-based method was used for wheat germ. ESPRESSO presents two prediction methods: (i) sequence/predicted structural property-based and (ii) sequence pattern-based method. The second method enables users to mutate candidate predicted regions for improving expression and/or solubility. For this method, the scoring function is calculated as the difference between frequencies of sequence patterns in positive (soluble) and negative (insoluble) datasets and is presented as a normalized adimensional 0 to 1 value with an associated *p*-value for each score. In this way, predictions can discriminate soluble from insoluble regions or motifs. For insoluble regions, ESPRESSO suggests point mutations for enhancing solubility and so, it can be used for rational design of insoluble proteins. An example of this use is the selection for experimental testing of different anthranilate phosphoribosyltransferase (AnPRT) protein variants with potentially high solubility [199].

#### 5.1.2. Structure-Based Analysis

The CamSol [200] web server consists of three algorithms that can be used individually for specific tasks or together to rationally design protein variants with enhanced solubility: (i) a sequence-based predictor of intrinsic solubility profiles and solubility scores, (ii) an algorithm exploiting knowledge of the native structure to perform structural corrections to the intrinsic solubility profile, and (iii) an algorithm analyzing the solubility profile to identify suitable sites for amino acids substitution or insertion. Increasingly negative profiles represent increasingly insoluble regions, while positive profiles represent increasingly soluble ones. The first algorithm can be used to screen protein-variant libraries for enhanced solubility. When combining the three algorithms, CamSol performs a systematic screening of thousands of substitutions and insertions while preserving fundamental properties to identify the most soluble variant. Interestingly, CamSol accepts (i) low-resolution and homology modeling-derived structures and (ii) a list of non-mutable residues (catalytic or structurally important). Pascal et al. used the CamSol web server for the assessment of the solubility profile and posterior rational design of a plant rhabdovirus glycoprotein for the production of immunoreactive murine anti-sera [201]. Lettuce necrotic yellow virus (LNYV) rhabdovirus glycoprotein native signal peptide was substituted with that of Rabies virus glycoprotein based on CamSol solubility predictions. His_6_ and FLAG-tags were added at N and C-termini, respectively, which were also predicted to enhance solubility. In this case, an increased glycoprotein solubility had been previously related with higher expression yields. Another example is the rational design of β-2 microglobulin (β-2m) mutants with reduced aggregation propensity, which is calculated as the inverse of CamSol solubility score. Selected substitution sites were mutated to all possible amino acids. The results identified the V85E mutation to be aggregation-resistant but with reduced thermostability (with a Tm value decreased by about 3 °C relative to WT). More recently, CamSol was used for the solubility prediction of a multi-epitope vaccine against SARS-CoV-2 developed by Martin and Cheng [203].

AGGRESCAN3D [204,205] (A3D) is a structure-based solubility prediction method, developed from the combination sequence-based AGGRESCAN [214] residue aggregation propensity and structural information. A3D allows for the detection of APRs while incorporating a mutation module that allows the design of proteins with increased solubility by mutating the detected aggregation-prone residues or their surroundings. The aggregation propensity is calculated for spherical regions centered on every residue Cα carbon. Moreover, A3D presents a ‘Dynamic Mode’ to analyze the impact of structural fluctuations on the aggregation propensity by minimizing the structure using FoldX, followed by CABS-flex [215] simulation of protein structure flexibility. The resulting trajectory is automatically processed to provide a set of protein models (in an all-atom resolution) reflecting the most dominant structural fluctuations in the near-native ensemble. Recently, an extension of A3D was released as a 2.0 version with a larger analysis range for proteins of up to 4000 residues long, a feature for simultaneous prediction of changes in protein solubility and stability and an ‘automated mutations’ tool for suggesting protein variants with optimized solubility.

A more recent solubility prediction tool is AggScore [206]. The algorithm is entirely based on a three-dimensional structure and is able to identify APRs by quantifying the energetic contribution of each residue to respective hydrophobic and electrostatic surface patches. After the identification of APRs, these surface patches are further classified into three categories, based on their surface potential values: hydrophobic, positive and negative regions. The hydrophobic potential is calculated for each atom based on logP parameters projected onto the interaction surface. On the other hand, positive and negative hydrophilic APRs are calculated using atom partial charges for the surface projections. The scoring function for aggregation propensity (AggScore) calculates the intensity and relative orientation of these respective APRs as the sum of the aggregation propensity values at each residue.

#### 5.1.3. Machine Learning

ML methods have historically been the most widely used for predicting protein solubility [94,216], mainly for the enhancement of recombinant overexpression in *E. coli* [216]. PON-sol [207] web server is a 2-layer Random Forest (ML) predictor that classifies mutant protein variants into three classes: increased, decreased and no effect on solubility. The software performance was tested and compared against CamSol and OptoSolmut [208], where it showed higher predictive accuracy using blind test, with 43% of correct predictions against 35% and 28% for CamSol and OptoSolmut, respectively, although the dataset consisted of less than 400 protein variants. PON-sol was applied to Interleukin-1β as a case of example, resulting in 1030 mutations predicted to increase solubility from a total set of 2907 variants (35.4%). Among these, seven positions were characterized as hotspots contributing the most to increase the solubility predicted score when mutated.

OptSolmut [208] is an ML-derived scoring function capable of measuring the “degree of buriedness” for three body contacts under the framework of Delaunay Tesselation (DT) (a geometry–based construct defining clusters of nearest neighboring points or four body contacts. It was used for the successful identification of mutants improving stability and enzyme reactivity. The degree of buriedness is a coarse-grained estimation of surface exposure with no measure of surface areas allowing for a better definition of neighboring residues, with the assumption that solubility is predominantly a surface property. The scoffing function for solubility prediction is based on measuring the frequencies of amino acid triplets presenting low ‘buriedness degree’, meaning that these triplets are located predominantly at the protein surface. The three-body contacts or triangles are classified as buried when forming part of two tetrahedrons in DT. The total score of a protein structure conformation is the sum of individual scores of amino acid triplets. Importantly, this scoring function based on groups of surface residues allows for the prediction of single and multiple-point mutational effects on solubility in a unified way, in contrast with most solubility prediction tools. The scoring function was trained using the ML Linear Programming (LP) approach for binary classification of amino acid triangles as buried and non-buried. The training was carried out using a dataset consisting of 137 single- and multiple-point mutants for changes in solubility extracted from the literature. Cross-validation studies were compared with two classification methods, SVM and Lasso, where OptSolmut outperformed both with an 81% overall accuracy. However, results should be taken with care given the small dataset used for validation.

One of the most recently developed ML tools for the prediction of solubility enhancement is Cordax [209], a structure-based machine learning approach that explores sequence determinants of amyloid propensity. Cordax explores amyloid sequence beyond the identification of APRs. First, a curated dataset was built consisting of 78 short-segment fibril core high-resolution structures from PDB, which were grouped into distinct classes based on topology and their overall structural properties. Cordax followed the same initial approximation as used in GAP, dividing the amino acid sequences into hexapeptides for the training dataset, yielding 179 peptide fragment structures. Then, amyloid interaction interfaces were analyzed in detail following energy refinement by the FoldX force field. Free energies calculated with Foldx were used to train a logistic regression model with binary classification. The prediction output of Cordax is multiple: First, there is the prediction from the logistic regression whether the segment is an amyloid core sequence. Second, for the sequences classified as amyloid core-forming, the most likely amyloid core model is provided. Hexapeptides presenting scores equal to or above the aggregation propensity threshold (0.71) are considered APRs. In this way, Cordax enables the prediction of APRs with different feature properties such as high solubility, high net charge, surface exposure in protein native folds, composition similarity to phase transition sequences and disorder or α-helix propensity (conformational switches). Cross-validation accuracy studies for Cordax were compared with different methods for amyloid aggregation propensity prediction such as TANGO, PASTA, AGGRESCAN and GAP, previously explained in this section. The receiver operating characteristic (ROC) curves generated showed that Cordax performance is the best among all other predictors tested, with an accuracy of 0.81. This high accuracy in identifying amyloid fibril-forming regions can be explained, as Cordax is able to detect APRs not presenting typical sequence propensities detected by sequence-based predictors, such as hydrophobicity or β-structure tendency. Cordax accuracy was further validated by synthesizing a subset of 96 peptides from detected APRs of the protein initial training dataset with more than half (55.3%) being predicted specifically by Cordax. Apart from a large cluster corresponding to sequences found in the hydrophobic core of globular proteins, Cordax also found clusters corresponding to surface-exposed amyloid sequences, small aliphatic functional amyloids, N/Q/Y prions, strongly helical and intrinsically disordered sequences which could be compatible with liquid–liquid phase responsive sequences.

#### 5.1.4. Hybrid Approaches

SolubiS [210] is a hybrid method for the prediction of mutants reducing aggregation tendency implemented as a plugin in YASARA [217] software (molecular graphics, modeling and simulation program). It combines TANGO [179] (detailed in the sequence-based section) and FoldX [218] tools to guide the design of aggregation-resistant protein sequences. The implementation of FoldX allows for the calculation of the overall energetic effects on protein stability for every APR detected. Importantly, SolubiS presents functionality that allows mutating one or several residues to calculate the effects (increase or decrease) over aggregation tendency presenting results on an easy-to-interpret ‘stretch-plot’. It is worth noting that, while TANGO can detect multiple APRs in a protein, SolubiS score is able to discriminate which of these APRs are solvent-accessible, determining aggregation propensity and thus, mutation targetable. In addition, SolubiS allows for the evaluation of the influence of temperature, ionic strength and pH on the aggregation prediction. SolubiS methodology has successfully been used for decreasing the aggregation propensity of human lysosomal hydrolase α-galactosidase (α-Gal) and protective antigen (PA) of *Bacillus anthracis* (anthrax vaccine formulation) [180], as well as for the engineering of human antibody variable domains with increased aggregation resistance [182].

### 5.2. Examples of Protein Solubility Enhancement Using Computational Tools

#### 5.2.1. Computational Design and Biophysical Characterization of Aggregation-Resistant Point Mutations for Human γD Crystallin [219]

Human γD crystallin is a stable protein expressed in the eye and responsible for lens transparency. However, this protein is susceptible to aggregation during the refolding process. RosettaDesign in combination with aggregation-propensity calculations (AGGRESCAN, PASTA, and TANGO) was used to predict mutants that are resistant to aggregation by measuring the effect of protein mutations on relative unfolding free energies (ΔΔG_un_) and intrinsic aggregation propensity (IAP). Despite being the least conformationally stable mutation, S130P was the most resistant to aggregation variant of Human γD crystallin, indicating a significant decrease in its aggregation propensity compared to WT.

#### 5.2.2. Prediction of Hotspots for the Reduction of Aggregation Propensity of Human α-Galactosidase and Protective Antigen of Bacillus Anthracis

Schymkowitz et al. [180] used SolubiS methodology to decrease the aggregation propensity of human lysosomal hydrolase α-galactosidase (α-Gal) and protective antigen (PA) of *Bacillus anthracis* (anthrax vaccine formulation). α-Gal deficiency causes Fabry disease, which can be treated by enzyme replacement therapy. α-Gal was engineered using SolubiS methodology to decrease its aggregation tendency by suppressing APRs. Mutational scanning of gatekeeper residues (charged or proline residues reducing aggregation [220]) resulted in the identification of A348R and A368P stabilizing APR without lowering intrinsic aggregation. In addition, an exhaustive mutation scan was performed looking for enhanced thermodynamic stability. The results showed S405L tightening the interaction of the edge β-strand. Overall, single mutants showed an experimental solubility increase of up to 80–90% compared with 70% of WT protein. Moreover, double and triple mutants showed a decrease in insoluble fraction by Western blot analysis. On the other hand, the rational design of aggregation-resistance PA using Solubis demonstrated that domain 3 is responsible for in vitro aggregation by identifying the double mutant T576E/S559L presenting improved thermodynamic stability and increased aggregation resistance at 40 °C. Overall, Schymkowitz et al.’s [180] research demonstrated that alteration in the number of APRs has a direct correlation on protein solubility and abundance. Another example of SolubiS use is the engineering of human antibody variable domains with increased aggregation resistance [182]. In this study, APRs present at complementary determining regions (CDRs) of monoclonal antibodies were demonstrated to determine the aggregation behavior under mild temperatures. SolubiS was applied to vascular endothelial growth factor (VEGF) for the rational design of mutations targeting APRs resulting in the identification of different mutant antibodies with improved aggregation resistance under temperature stress.

## 6. Concluding Remarks and Outlook

This review has presented an overview of the computational methods used to enhance key biocatalytic properties of enzymes, focusing specifically on stability, solubility, substrate specificity, and catalytic efficiency. These methods have proven to be robust and versatile, enabling detailed exploration of enzyme behavior and properties at atomic and molecular levels. They were grouped by the catalytic property they optimize with the aim of facilitating the selection by non-expert users to facilitate theoretical to experimental workflows for enzyme optimization.

Despite the advancements in the field, challenges remain. The accuracy of the current computational predictions often relies on the availability of high-quality structural and mechanistic data, which is not always accessible. Additionally, modeling enzyme behavior under non-standard or highly variable conditions, such as extreme temperatures or complex environments, continues to be a limitation. Improvements in scoring functions, particularly those capable of capturing dynamic properties like allosteric effects or solvent interactions, are essential for furthering the utility of these tools. Moreover, greater integration of experimental data into computational frameworks will enhance the reliability of the predictions.

This review deliberately excluded discussions on deep learning methods and de novo protein design, both of which have been reviewed extensively elsewhere. These areas represent a paradigm shift in computational enzyme design. Deep learning approaches, in particular, have demonstrated the ability to process large datasets, uncover hidden patterns in protein sequences, and generate innovative designs beyond the reach of traditional methods. Similarly, de novo design allows for the creation of entirely new enzymes, expanding the boundaries of what is possible in protein engineering.

Looking ahead, the future of computational enzyme design lies in the integration of traditional molecular modeling approaches with the transformative potential of artificial intelligence (AI). Combining the mechanistic insight and interpretability of molecular modeling with the predictive power and data-driven nature of AI offers an unprecedented opportunity to overcome current limitations. For instance, AI-driven tools could refine molecular dynamics simulations by identifying key conformational changes or accelerate sequence space exploration by prioritizing mutations with high potential for success.

Additionally, hybrid approaches that leverage both rational design principles and machine learning models could improve the accuracy of scoring functions, allowing for better prediction of enzymatic properties under diverse conditions. These synergies could also facilitate the design of multi-functional enzymes or enzymes tailored to highly specific industrial applications, such as green chemistry, pharmaceutical synthesis, or bioenergy production.

In conclusion, while traditional computational methods remain a cornerstone of enzyme engineering, the rapid evolution of AI and deep learning is set to redefine the field. The integration of these approaches will enable more efficient, accurate, and innovative enzyme designs, unlocking new possibilities in biotechnology.

## Figures and Tables

**Figure 1 ijms-26-00980-f001:**
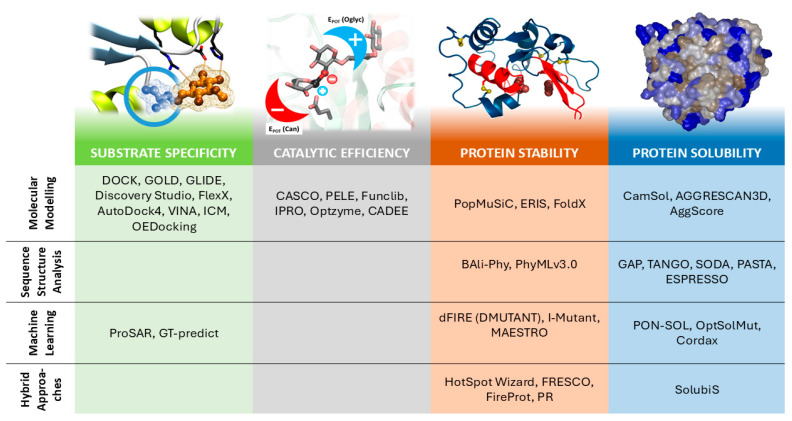
Most common computational tools for engineering biocatalytic properties of enzymes. Tools are grouped by the property they target and by the theoretical framework they rely on.

**Table 1 ijms-26-00980-t001:** A list of computational tools commonly used to assist in the engineering of enzyme–substrate recognition, and some applications.

Methodology	Computational Tool *	Application Examples
**Virtual** **Docking**	DOCK [7]	Glycolaldehyde synthase and phospoketolase [8], Acetohydroxyacid synthase [9], β-xylosidase and β-glucosidase [10]
	GOLD [11]	Cytochrome P450 stereo-selectivity [12], Lipase [13], Metallo-β-lactamase [14], Transglycosidase [15], GPCR [16]
	Discovery Studio [17]	UDP-glucosyltransferase [18], Cephalosporin C acylase [19]
	GLIDE [20]	ω-Aminotransferase [21], Alcohol dehydrogenase [22], D-Lactate dehydrogenase [23], Cutinase [24]
	FlexX [25]	N-Myristoyltransferases [26], Serine protease [27], Laccase and Azoreductase [28]
	AutoDock4 [29]/VINA [30]	Azoreductase [31], Lipases [32,33,34], Serine protease [35]
	ICM [36]	MOR42-3 receptor [37], Alcohol acyl transferase [38], β-Lactamase [39]
	OEDocking [40]	Butyrylcholinesterase [41]
**Machine** **Learning**	ProSAR [42]	Halohydrin dehalogenase [43] Dextransucrase and Cytochrome P450 [44] Transglucosylase [45]
	GT-predict [46]	Glycosyl Transferase Superfamily 1 [46]

* Current URL links (accessed on 30 December 2024): DOCK: http://dock.compbio.ucsf.edu/index.html; GOLD: https://www.ccdc.cam.ac.uk/solutions/software/gold/; Discovery Studio: https://www.3ds.com/products/biovia/discovery-studio; GLIDE: https://www.schrodinger.com/platform/products/glide/; FlexX: https://www.biosolveit.de/products/#FlexX; AutoDock4: https://autodock.scripps.edu/; AutoDock VINA: https://vina.scripps.edu/; ICM: https://www.molsoft.com/docking.html; GT-predict: https://doi.org/10.5287/bodleian:zg5195kaE; OEDocking: https://www.eyesopen.com/oedocking.

**Table 2 ijms-26-00980-t002:** Scoring functions commonly used in virtual docking to evaluate protein–ligand binding affinities.

Theoretical Framework	Scoring Function
Physics-based	AMBER force field [51], Tripos force field [52], ECEPP/3 force field [53]
Empirical	Chemscore [54] and ChemPLP, FleX score [25], Surflex [55], Molegro Virtual Docker [56]
Semi-empirical	Goldscore [11], GlideScore [20], X-Score [57], AutoDock4.2 score [58], VINA score [30]
Knowledge-based	PMF [59] and PMF04 [60], Chemical Gaussian Overlay and Chemical Gaussian Tanimoto [40], Chemgauss [61], Drugscore [62]
Machine learning	RF-Score-v3 [63], RF-Score-VS [64], Δ_vina_RF_20_ [65], SVMGen [66], ID-Score [67], PLEIC-SVM [68], DDFA [69], BgN-Score/BsN-Score [70], NNScore 2.0 [71]

**Table 3 ijms-26-00980-t003:** A list of computational tools and protocols commonly used to assist in the engineering catalytic efficiency of enzymes, and selected applications.

Theoretical Framework	Computational Tool *	Application Examples
**Force fields**	CASCO [113,114]	Limonene epoxide hydrolase [115]
	Sherman (protocol) [21]	ω-aminotransferase [21]
	Warden (protocol) [106]	Transaminase [106]
	PELE [116,117]	Fungal laccase [104,105,111]
	FuncLib [118]	Acetyl-CoA synthases [118]
**Quantum** **Mechanics/** **Molecular** **Mechanics**	Houk (protocols) [119,120]	P450-monooxygenase [119] Dirhodium catalyst [120] Spiroligozymes [121]
	Cerqueira (protocol) [109]	L-asparaginase [109]
	Optzyme [122]	β-glucuronidase [122]
**Empirical** **Valence Bond**	CADEE [123]	Triosephosphate isomerase [123]
**Virtual Docking**	IPRO [124,125]	Acyl-ACP thioesterease [126]

* Current URL links (accessed on 30 December 2024): PELE: https://pele.bsc.es/pele.wt; FuncLib: https://ablift.weizmann.ac.il/step/fl_terms/; Optzyme: https://www.maranasgroup.com/submission/OptZyme.htm; CADEE: https://github.com/kamerlinlab/cadee; IPRO: https://www.maranasgroup.com/submission/ipro2014.htm.

**Table 4 ijms-26-00980-t004:** A list of computational tools and protocols commonly used to assist in the engineering of protein stability, and selected applications.

**Theoretical Framework**	**Computational Tool ***	**Application Examples**
**Phylogenetic Analysis**	BAli-Phy [131]	Adenylate kinase [132]
	PhyMLv3.0 [133]	Haloalkane dehalogenase [134]
**Molecular** **Modeling**	PopMuSiC (ProTherm) [135]	α-Amylase [136], Xylanase [137]
	ERIS [138]	DFPase [139]
	FoldX [140]	Pullulanase [141], Glucose oxidase [142]
**Knowledge-based**	dDFIRE (DMUTANT) [143,144]	Pullulanase [141]
**Machine** **Learning**	I-Mutant 3.0 [145]	Pullulanase [141]
	MAESTRO [146]	Phytase [147], Pollen allergen Phl-p-6 [148]
**Hybrid** **approaches**	HotSpot Wizard 3.0 [149]	Epimerase [150], Xylanase [151]
	FRESCO [152]	Limonene epoxide hydrolase [152], Glucose oxidase [142]
	FireProt [153]	Haloalkane dehalogenase [153]
	PROSS [154]	Chondroitinase ABC [155], hAChE [154]

* Current links (accessed on 30 December 2024): BAli-Phy: http://www.bali-phy.org/; PhyML: http://www.atgc-montpellier.fr/phyml/; PopMuSiC: https://bio.tools/popmusic; ERIS: http://eris.dokhlab.org; FoldX: http://foldxsuite.crg.eu/; dDFIRE: https://sparks-lab.org/downloads/; I-Mutant: https://bio.tools/i-mutant_suite; MAESTRO: https://pbwww.che.sbg.ac.at/; HotSpot Wizard: http://loschmidt.chemi.muni.cz/hotspotwizard/; FireProt: http://loschmidt.chemi.muni.cz/fireprotweb/; PROSS: http://pross.weizmann.ac.il/.

**Table 5 ijms-26-00980-t005:** A list of computational tools and protocols commonly used to assist in the improvement of protein solubility, and selected applications.

Theoretical Framework	Computational Tool *	Application Examples
**Sequence** **analysis**	GAP [188] (SAP)	Superoxide dismutase [189]
	TANGO [179]	Cyclic Peptides [190], SH2 domain [191], Mab1 [192]
	SODA [193]	CST complex subunit STN1 [194]
	PASTA 2.0 [195]	Triosephosphate isomerase [196], Helicase [197]
	ESPRESSO [198]	Phosphoribosyltransferase [199]
**Structure** **analysis**	CamSol [200]	Glycoprotein [201], β-2 microglobulin [202], COVID-19 vaccine [203]
	AGGRESCAN3D 2.0 [204,205]	GFP and Human germline antibody [204,205]
	AggScore [206]	Amyloid precursor protein (APP) and clinical-stage antibodies [206]
**Machine** **Learning**	PON-sol [207]	Human interleukin-1β [207]
	OptSolMut [208]	137 protein solubility datasets [208]
	Cordax [209]	APR nucleators design [209]
**Hybrid**	SolubiS [210]	α-Galactosidase and Protective Antigen [180], Monoclonal antibodies [182]

* Current links (accessed on 30 December 2024): GAP: https://www.iitm.ac.in/bioinfo/GAP/; TANGO: https://tango.crg.es/; SODA: http://old.protein.bio.unipd.it/soda/; PASTA: http://old.protein.bio.unipd.it/pasta2/; CamSol: http://www-vendruscolo.ch.cam.ac.uk/camsolmethod.html; A3D2: https://biocomp.chem.uw.edu.pl/A3D2/; PON-sol: http://139.196.42.166:8010/PON-Sol2/; SOLUPORTMUT: https://loschmidt.chemi.muni.cz/soluprotmutdb/; Cordax: https://cordax.switchlab.org/; SolubiS: https://solubis.switchlab.org/.

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
