# Peer review of "A Practical Guide to Computational Tools for Engineering Biocatalytic Properties"

_ijms, 2025, doi:10.3390/ijms26030980_

Round 1

Reviewer 1 Report

Comments and Suggestions for Authors

In this manuscript, the authors summarized the bioinformatic tools for engineering of proteins and enzymes. This is a significant review for both fundamental research and practical application. Overall, these tools and methods have been widely used and achieve numerous engineered proteins. One of my concerns is that it is unclear what kinds of tools are more popular in use, because some tools are not well suitable to most of engineering cases. I suggest the authors supplement some summary and discussion for this issue.

Author Response

Comment 1: In this manuscript, the authors summarized the bioinformatic tools for engineering of proteins and enzymes. This is a significant review for both fundamental research and practical application. Overall, these tools and methods have been widely used and achieve numerous engineered proteins.

Response 1: We acknowledge the referee for positively rising the significance of our review in both fundamental and applied research.

Comment 2: One of my concerns is that it is unclear what kinds of tools are more popular in use, because some tools are not well suitable to most of engineering cases. I suggest the authors supplement some summary and discussion for this issue.

Response 2: We agree with the referee that not all tools are suitable for every protein engineering scenario. That’s why we have structured the review focusing on the biocatalytic properties. For each specific protein engineering scenario, we aim at providing diverse computational approaches with different theoretical backgrounds. We prefer not to highlight some tools over the other, or easy to use software as this may introduce a bias from our personal experience. Rather, we have tried to indicate, throughout the text, those tools that are widely used by the community. This can be found, for example, in:

  • Page 3 (last paragraph): Molecular docking has been the most used computational method for improving protein-ligand binding affinity…
  • Page 4 (second paragraph): DOCK has been extensively used in enzyme design for different purposes,…
  • Page 4 (fourth paragraph): ICM has been widely used for virtual screening upon protein structures…
  • Page 5 (second paragraph): FlexX [25] is one of the most used docking software historically.
  • Page 5 (third paragraph): Other less extended virtual docking tools that implement empirical scoring func-tions are Surflex and MolDcok.
  • Page 5 (seventh paragraph): AutoDock4 [29] is the last version of AutoDock original software, the crown jewel among docking software with more than 10 thousand citations to date.
  • Page 12 (fourth paragraph): The IPRO [124,125,128] suite of programs has been extensively used for different enzyme redesign purposes.
  • Page 23 (last paragraph): ML methods have been historically the most widely used for predicting protein solubility…

Reviewer 2 Report

Comments and Suggestions for Authors

This is a well-written guide that addresses the state-of-the-art computational tools for enzyme design. The structure of the review is laid out in a coherent fashion, and I simply cannot find any mistakes or inadequacies within this manuscript. I would love to see this review published as is.

I still standby my original review, and I usually give very in-depth comments when I find methodological/logical errors and even something as simple as grammatical errors. However, this manuscript must have been proofread many times such that I cannot find anything I can nitpick about. That being said, I can give a more formal review to satisfy your assessments, but the conclusions do not change.

In this review, the authors have detailed computational tools that are developed for re-designing (or improving) naturally occurring enzymes with the criteria of desired properties, including protein-ligand affinity, catalytic efficiency, thermostability, and solubility. These are all very important properties, especially for industrial applications of enzymatic catalysis, so I find this review very timely. As experimentalists, we do not often find easy ways to address these properties without extensive trial and error, while computational approaches can greatly reduce time and efforts in this regard. The authors have laid out different approaches to tackle each category of properties, along with their up-to-date applications. What I find the most commendable in this review is that while the hottest trend of enzyme development has been focusing on the use of machine learning methods or de novo enzyme design (see 2024 Nobel Prizes), the authors have refrained from including these topics with clear "caveats" noted and instead have written this review which will appeal to the more general public (your everyday enzyme researcher). I guess the only thing I can nitpick about is that there is a lack of illustration throughout the manuscript, which is rather unusual for topics on enzyme design. I suppose that this is because it is difficult to obtain copyright to reproduce figures from journals, but I actually don't find this "supposed shortcoming" hindering the readability of the manuscript given the authors' excellent writing abilities. In fact, the tables have already compiled the relevant references and methods mentioned in the review, so I do not think further improvements are necessary.

Author Response

Comment 1: This is a well-written guide that addresses the state-of-the-art computational tools for enzyme design. The structure of the review is laid out in a coherent fashion, and I simply cannot find any mistakes or inadequacies within this manuscript. I would love to see this review published as is.

I still standby my original review, and I usually give very in-depth comments when I find methodological/logical errors and even something as simple as grammatical errors. However, this manuscript must have been proofread many times such that I cannot find anything I can nitpick about.

 Response 1: We deeply appreciate the positive comment of the referee on the quality and coherence of our review manuscript.

Comment 2: That being said, I can give a more formal review to satisfy your assessments, but the conclusions do not change. In this review, the authors have detailed computational tools that are developed for re-designing (or improving) naturally occurring enzymes with the criteria of desired properties, including protein-ligand affinity, catalytic efficiency, thermostability, and solubility. These are all very important properties, especially for industrial applications of enzymatic catalysis, so I find this review very timely.

Response 2: We acknowledge the referee for his comment on the timely relevance of our review manuscript.

Comment 3: As experimentalists, we do not often find easy ways to address these properties without extensive trial and error, while computational approaches can greatly reduce time and efforts in this regard. The authors have laid out different approaches to tackle each category of properties, along with their up-to-date applications. What I find the most commendable in this review is that while the hottest trend of enzyme development has been focusing on the use of machine learning methods or de novo enzyme design (see 2024 Nobel Prizes), the authors have refrained from including these topics with clear "caveats" noted and instead have written this review which will appeal to the more general public (your everyday enzyme researcher).

Response 3: We agree with the referee’s perspective on our manuscript review as an experimentalist. We were aware of that when planning the review. Our intention is to provide a short hand to the most commonly used computational tools in the field, with easy access to everybody. That’s why we also provide updated current web links to the tools at the bottom of each table.

Comment 4: I guess the only thing I can nitpick about is that there is a lack of illustration throughout the manuscript, which is rather unusual for topics on enzyme design. I suppose that this is because it is difficult to obtain copyright to reproduce figures from journals, but I actually don't find this "supposed shortcoming" hindering the readability of the manuscript given the authors' excellent writing abilities. In fact, the tables have already compiled the relevant references and methods mentioned in the review, so I do not think further improvements are necessary.

Response 4: We agree with the referee’s comment on the lack of illustration in the review manuscript. The reason for that is that we didn’t want to reproduce illustrations from previous publications, as the main objective of the review is to provide a practical list of available tools, rather than discussing in detail the results obtained with them. Nevertheless, we agree the review manuscript may benefit with some graphical illustration of the contents. We have included a figure summarizing the contents of the review, providing an easy-to-visualize list of computational tools grouped by biocatalytic property. This figure is now Figure 1, and it is located at the end of the introduction (page 2).